# Acoustic wave response to groove arrays in model ears

**Brian W. Keeley** [1]☉¤*, **Annika T. H. Keeley** [2]☉¤

**1** Miridae, Sacramento, California, United States of America, **2** Delta Stewardship Council, Sacramento, California, United States of America

☉ These authors contributed equally to this work.
¤ Current address: Davis, California, United States of America
* bkeeley62@yahoo.com

**Data Availability Statement:** The raw data and processed data are included in the submission.

**Funding:** The author(s) received no specific funding for this work.

## Abstract

Many mammals and some owls have parallel grooved structures associated with auditory structures that may be exploiting acoustic products generated by groove arrays. To test the hypothesis that morphological structures in the ear can manipulate acoustic information, we expose a series of similar-sized models with and without groove arrays to different sounds in identical conditions and compare their amplitude and frequency responses. We demonstrate how two different acoustic signals are uniquely influenced by the models. Depending on multiple factors (i.e., array characteristics, acoustic signal used, and distance from source) the presence of an array can increase the signal strength of select spectral components when compared to a model with no array. With few exceptions, the models with arrays increased the total amplitude of acoustic signals over that of the smooth model at all distances we tested up to 160 centimeters. We conclude that the ability to uniquely alter the signal based on an array's characteristics is evolutionarily beneficial and supports the concept that different species have different array configurations associated with their biological needs.

## 1. Introduction

Animals that are completely or partially nocturnal have adaptations that allow them to navigate in low light conditions [1]. Echolocating bats use one of the more advanced adaptations for navigating in low light conditions by listening for echoes produced from vocalizations [2,3]. The pinnae of most bat species have a series of equally spaced grooves that form an array [4]. Similar arrays also are present in nonecholocating bat species, many species of mammals and some owls. These arrays are located on structures in the path of acoustic waves entering the auditory canal. Specialized features such as enlarged pinnae in mammals and facial discs in owls assist the listening process [1,5]. Studies have shown that structural features within mammalian pinnae (e.g., tragus, lobes, folds) and the pre-aural skin flap in the facial disc of owls (operculum) are known to cause a range of acoustic effects [5–7]. We hypothesize that the groove arrays linked to aural features are causing beneficial acoustic effects.

**Competing interests:** The authors have declared that no competing interests exist.

Many animals, including a small number of bat species, have pinnae with a smooth reflective surface. When waves encounter a reflective surface, the angle of reflection from a smooth flat surface is equal to the angle of incidence (Fig 1). The energy from an incoming sound wave will reflect in a predictable direction and pattern from the smooth surface in the pinna where the size of the wavelengths and the characteristics and size of the pinna structure will dictate the response. For example, when sound waves encounter curved surfaces, the wave energy will be scattered by a convex curvature or concentrated into a focal region by a concave curvature (Fig 1).

Similarly, when acoustic waves of appropiate size encounter a series of parallel grooves or slits (diffraction grating) they will behave in certain predictable patterns [8]. A diffraction grating acts like a "super prism" that is a highly efficient tool not only for separating different spectral wavelengths but also for creating a framework to cause constructive interference which combines the amplitudes of similar frequencies. When waves encounter an object they will bend around it, termed diffraction. Wavefronts containing multiple frequencies that diffract from individual grooves bend at different rates because the different wavelengths cause them to separate and travel at different angles (Fig 2A). According to the Huygens-Fresnel principle, wavelets emerging from the grating's periodic structure causes the emerging wavelets to expand outward from measured locations creating organized patterns that causes both constructive and destructive interference [8] (Fig 2B) where the first order has the strongest amplitude.

Because the wavelets are generated from the same wavefront, they are identical in length and amplitude. When two waves of identical length and amplitude meet and are completely in phase, they undergo fully constructive interference or when 180 degrees out of phase they undergo fully destructive interference. Fully constructive interference will double the amplitude when the waves join, whereas fully destructive interference causes complete cancellation resulting in the amplitude dropping to zero (Fig 3). Changes to wave amplitude caused by interference are dependent on the direction of travel, the wavelengths involved, and whether the waves are completely or partially in phase with each other. Pure constructive or destructive wave interference likely occurs rarely outside of controlled settings because it requires

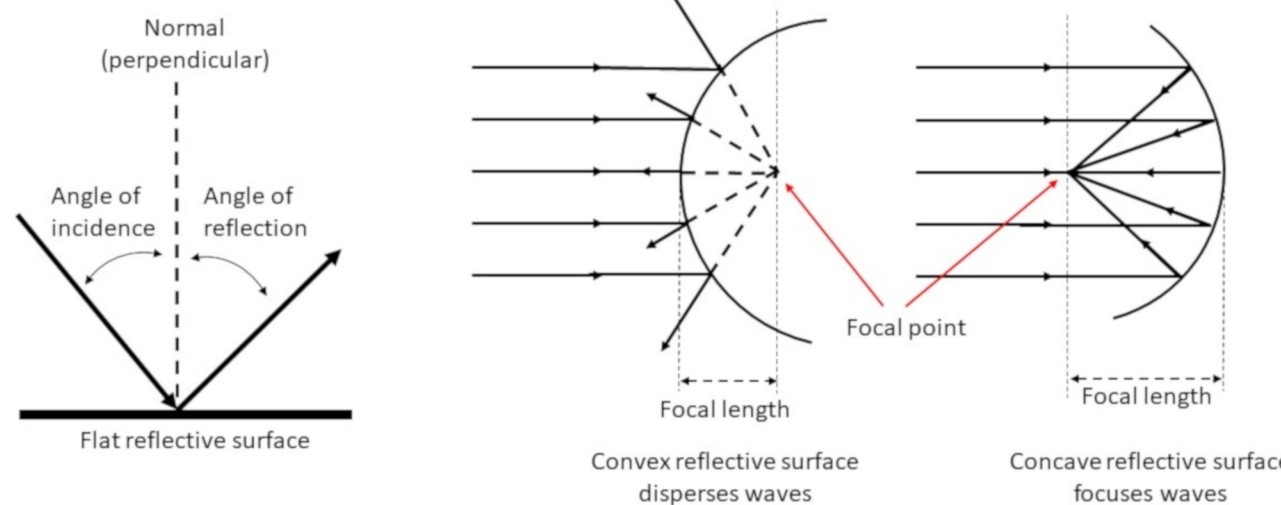

**Fig 1. Sound waves have predictable behaviors when they encounter flat, convex, or concave reflective surfaces.** The angle of incidence equals the angle of deflection [8].

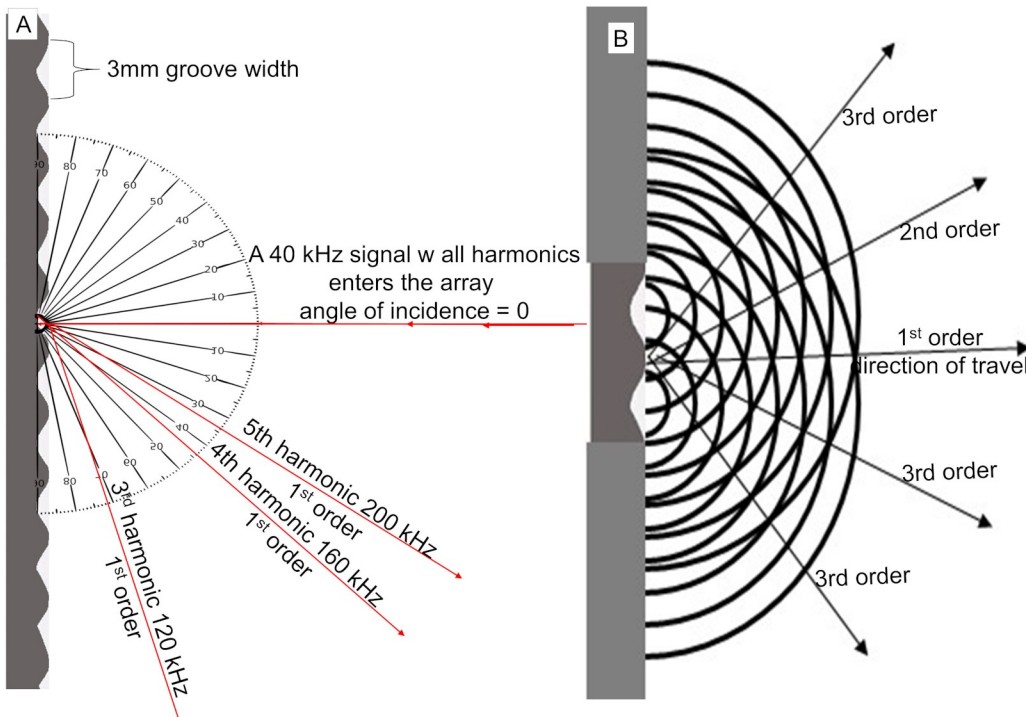

**Fig 2. Sound waves response to groove arrays.** (A) The direction of travel varies for different wavelengths. When a 40kHz harmonic series contacts a 3mm width groove array, the harmonics have different 1st order exit angles which can be calculated in relation to the 40 kHz fundamental and the angle of incidence is normal (= 0˚). Because the wavelengths of the first two harmonics (40 and 80 kHz wavelengths = 8.575mm and 4.2875mm) are too large to diffract from a 3mm width groove, diffraction occurs for all harmonics above the 3rd harmonic out of each groove. For clarity we only show the response from a single groove, but this would result from every groove. (B) Waves expanding from the grooves contact each other in organized patterns. As the number of grooves increases these patterns multiply. For clarity we show the pattern of interaction from just two grooves.

precisely aligned identical waves. The groove series creates a diffraction grating that provides the controlled setting where emerging wavelets can interact to cause the fully constructive and destructive interference patterns (Fig 3).

Controlling the angle of entry (angle of incidence) into the array by adjusting the head or moving the pinnae would ensure a wavefront contacts the array simultaneously, which would help to evenly distribute the response and synchronize the interference patterns and phase relationships of all frequencies within a broadband wavefront and more efficiently maximize energy retention. Combining the binaural information from the arrays from the two separate pinnae would help to triangulate and pinpoint the source location with the precision improving as the frequencies increase [9,10]. Likewise, according to wave behavior principles, the array characteristics (separation widths and/or groove number) determine which wavelengths can interact and the intensity of their response [8].

Most bat species (Fig 4A), a variety of other mammals [4] and some owls (Fig 4B–4D) have ridges that form a series of parallel grooves associated with the auricular opening that may assist the auditory process. In bats (and possibly other mammals with groove arrays), the grooves are formed from innervated muscles [11] whereas the arrays in owls appear to form from where feather quills anchor into the skin or from the alignment of naked quill shafts which would form slits that are also known to affect acoustic waves (Fig 4B–4D). Some members of both owl families (Tytonidae and Strigidae) have a facial disc and in some there is a

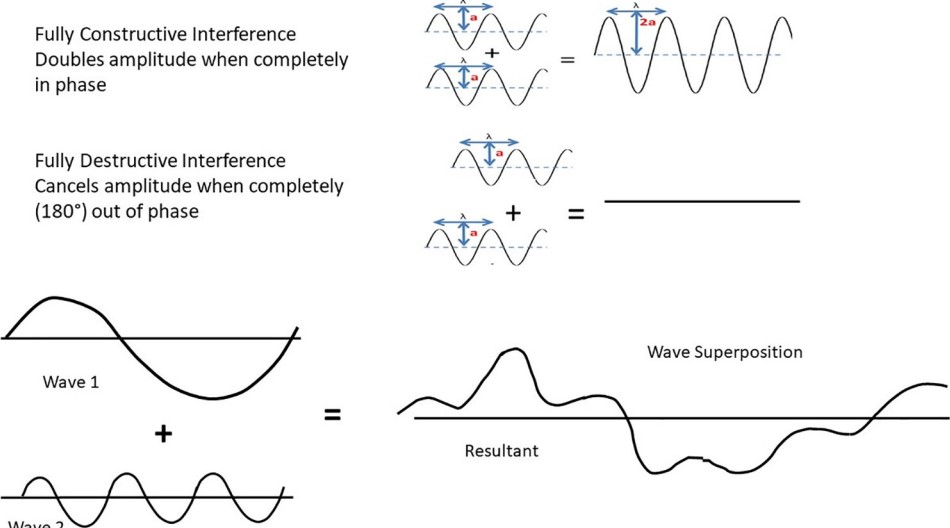

**Fig 3. Wave interference diagrams.** Two equal wavelengths of equal amplitude, will either double the amplitude during fully constructive interference when they are exactly in phase or completely cancel during fully destructive interference when exactly out of phase. Superposition of non-identical waves exhibits irregular patterns integrating both constructive and destructive interference.

mobile pre-aural flap called an operculum that has been observed to influence acoustic information [5,12]. Some owls in these families have quill array formations located either on the operculum or bordering the ear opening (or both) that are likely to affect acoustic waves in the same way that we are proposing the groove arrays affect acoustic information in mammals. Groove array characteristics, especially the number of grooves and the separation widths, can vary considerably between and within species [4] (Fig 4A).

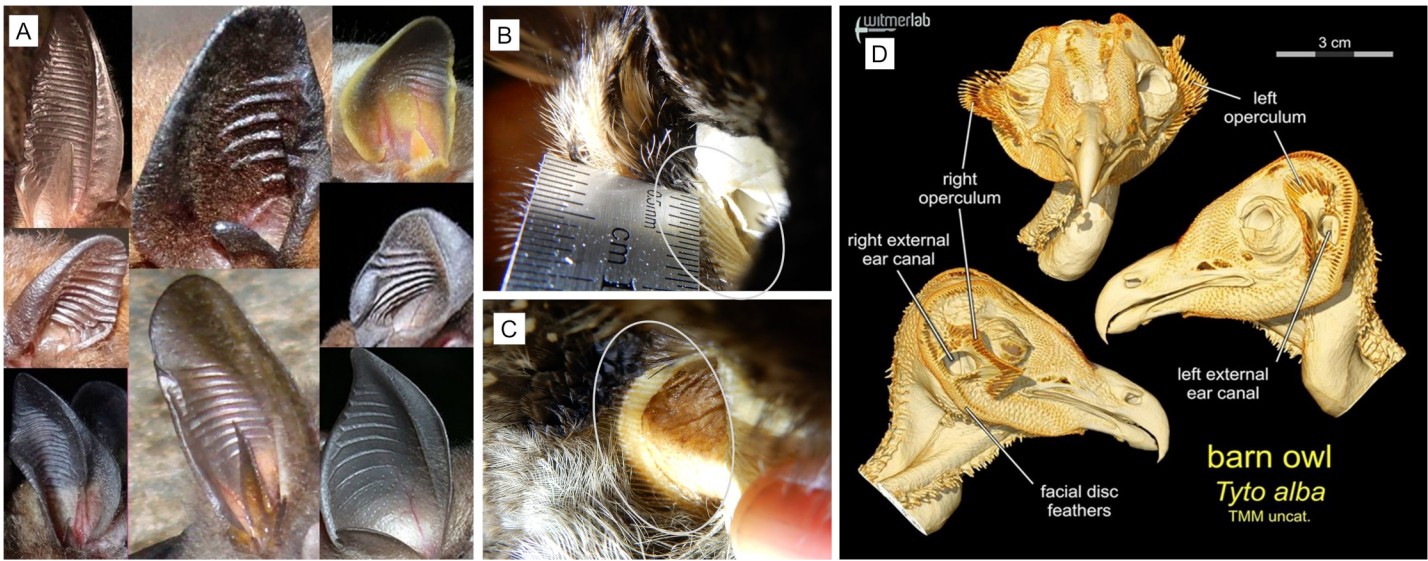

**Fig 4. Groove array variations in the ears of bats and owls.** (A) Groove array variations in bat pinnae. Owl museum specimen photos show periodic structures formed where quills anchor into the skin bordering the auditory openings for (B) Long-eared Owl (Asio otus), (C) Great Grey Owl (Strix nebulosa) and a computer rendering of (D) Barn Owl (Tyto alba) where feather shafts on the operculum and quill shafts bordering the auricular opening create periodic array structures.

In some bat species the grooves can be observed to participate in folding the pinnae, especially those with large pinnae and many grooves (*e.g.*, *Corynorhinus ssp.* and *Plecotus ssp.*). However, other species with groove arrays in their pinnae do not appear to be able to or need folding (e.g., *Eumops spp*, *Artibeus gnomus*) and sometimes the arrays (or portions of it) are located in regions that do not appear to be able to participate in folding. Therefore, folding may be a consequential and not a causal role of groove arrays.

Studies suggest that parallel ridges associated with the auricular opening of animals should create acoustic diffraction effects [13,14], and that variations in the configuration of the groove arrays may be linked to the foraging characteristics of bats [4]. Because nocturnal animals rely heavily on acoustic information for survival to navigate and forage, we hypothesize that the arrays are enhancing acoustic information. To test this hypothesis, we manufactured a series of pinna models that differ only in groove array configuration and used them to conduct a series of acoustic tests.

We test 2 hypotheses:

1. The presence of a groove array will provide different acoustic information than that of a smooth surface.

2. Based on the principles of diffraction, the presence of a groove array will alter the acoustic signal based on the array's characteristics.

   a. Amplitude will increase as the number of grooves increase.

   b. Wave lengths that correspond to specific groove separation widths will have a marked increase in amplitudes.

## 2. Materials and methods

We compared the acoustic responses of 10 pinna-shaped models using two sound sources at distances up to 160 cm.

### 2.1 Pinna models

We designed a set of pinna models based on acoustic principles, a review of photographs of groove-bearing pinnae, the literature [15], and 3D printing capabilities (Fig 5, Table 1). The models have identical dimensions of 28 mm x 44 mm but differ in the number of array grooves (0, 5, 10, or 20) and separation widths (1 mm, 1.715 mm, 2.858 mm and 3 mm; Table 1). The 3D printing lab stated that for the scale of these models the printer is accurate to 0.001 mm. The separation widths were selected to match the harmonics of 40 kHz wavelengths (see Table 1). Although the individual pinnae of bats are often asymmetrical, we created symmetrical pinna models to enable us to receive a balanced response from each model and to simplify our ability to compare, interpret and present the results. The models have a 4 mm tubular opening centered at the bottom specifically designed to snugly slip over a Pettersson M500 ultrasound microphone which has a 4 mm diameter x 6 mm long tubular pedestal that extends the microphone element out from the housing (Fig 5). When a pinna model is seated on the microphone housing tube, the microphone element fits level with the curved floor of the model.

### 2.2 Sound sources

We directed two different types of sounds towards the pinna models to test the effect of the groove arrays when contacted by: (1) a 40 kHz continuous tone square wave (and its

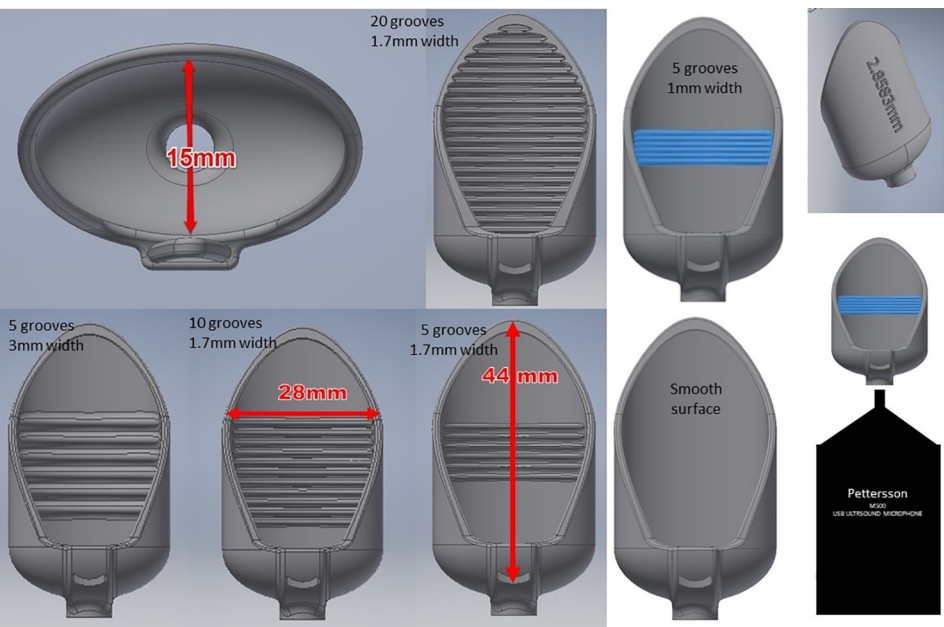

**Fig 5. Examples of pinna model variations and dimensions with microphone set up.**

harmonics) and (2) a manufactured prey noise (a broad band frequency spectrum fully described in [16]). The 40 kHz sound waves are contained in many bat calls [17], and the sounds created by prey items (e.g., crunching leaves) are monitored by passively listening species [16,18]. During the acoustic tests, all models were positioned so that the arriving sound was perpendicular or vertically orthogonal to the surface of the model which is labeled as "normal" when describing the angle of incidence contacting an array.

For comparing the influence to the signal by changing a model's characteristics, we needed a signal that contained multi-frequencies with consistent amplitude. We generated a 40 kHz continuous square wave of consistent amplitude using an ultrasound emitter ('BatChirp', built by Tony Messina). The square wave signal generates harmonics with emphasis on the odd harmonics. The harmonics generated are the 1st harmonic at 40 kHz (fundamental), the 2nd harmonic at 80 kHz, the 3rd harmonic at 120 kHz, the 4th harmonic at 160 kHz, and the 5th harmonic at 200 kHz. The harmonics allow evaluation of the influence of groove separation width variations (3rd hypothesis) and groove numbers. With this multi-frequency signal, we are able to compare the arrays to the smooth model and the different array characteristics to each other.

**Table 1. Ten different pinna model configurations were tested.**

| Number of models (10 total) | Names | Groove # | Separation width | Frequencies corresponding to separation width[*] |
|---|---|---|---|---|
| 1 | smooth | 0 | Smooth surface | Any |
| 3 | 1mm5g 1mm10g 1mm20g | 5, 10, 20 | 1 mm | 343 kHz |
| 3 | 1.7mm5g 1.7mm10g 1.7mm20g | 5, 10, 20 | 1.715 mm | 200 kHz |
| 1 | 1.8mm5g | 5 | 2.858 mm | 120 kHz |
| 2 | 3mm5g 3mm10g | 5, 10 | 3 mm | 114 kHz |

[*]Derived from the formula $\lambda = c/f$ where $\lambda$ = wavelength, c = speed of sound (343m/s in dry air at 20°C) and f = frequency.

To mimic prey noises like rustling leaves, we modified a portable cassette player by attaching two sandpaper discs to the rotating spindles in such a way that the rough sides contacted each other once during each rotation. This apparatus produces consistent acoustic pulses that are identical in amplitude, pulse timing and contains a broadband frequency spectrum that mimics prey-generated sounds like those produced when an insect walks in dry leaves [16].

## 2.3 Recordings

In a quiet environment, we directed the two types of sounds towards each pinna model from a fixed location. We captured the sounds intercepted by the model with a microphone (Pettersson Elektronik M500) inserted into the bottom of the model. We used Batsound Touch® software (Pettersson Elektronic) with the sampling rate set at 500kHz for recording. The Pettersson M500 microphone is sensitive to frequencies up to 224 kHz which allowed us to examine responses up to the 5th harmonic (200 kHz) of a 40 kHz wave. The signal strength is recorded in decibels (dB). For ultrasound signals the decibel level is given as a negative value with the strongest signal intensities approaching human hearing thresholds which are indicated in graphs at zero. To maximize recording conditions for each test, we constructed a jig that, during every test, was holding the microphone fitted on the pinna models in the same orientation and location relative to the sound source. This jig enabled us to swap out the models without touching any other part of the setup. To account for variations between recordings, we made five replicate recordings for each sound source and each of the 10 models with the 40 kHz continuous tone and the prey generated sound sources positioned at distances of 10 cm, 20 cm, 40 cm, 80 cm, or 160 cm.

## 2.4 Analysis

We generated spectrograms in BatSound 4.4 (Pettersson Elektronik, Inc). We selected a typical sequence from spectrograms to generate power. Spectrograms illustrate the spectrum of frequencies of a signal as it varies with time with the intensity shown by varying color or brightness; power spectra plot signal power against frequencies between zero to 250 kHz. These data can be stored in ASCII files and exported as numerical values in decibels (dB) associated with each frequency in kilohertz (kHz) can be displayed and analyzed in Excel spreadsheets. We compared the signal responses by analyzing differences in decibel values assigned to the different frequencies.

To account for the microphone's performance limits, we cut off all data at 224 kHz. For the five recordings of each 40 kHz and prey sounds set-up, we averaged the resulting decibel values. In order to understand how the acoustic responses differ for each of the comparison tests we conducted, we need to be able to see or measure the differences. Using the amplitude values from the recordings we can measure both the overall quantity of energy intercepted and how it is distributed within the spectral envelope. We used two methods to compare data generated in each run. (1) Using the five replicates, we averaged the total amplitude values of each frequency (0–224 kHz), which enables us to compare differences between pinna models with respect to spectral distribution and total energy retained. (2) We generated frequency difference graphs by subtracting the mean amplitude values at each frequency. When the frequency values for two models are equal, then the difference is zero and when one of the models is greater than the other it can be seen when graphed. One concept to keep in mind while viewing the test results is that the amplitude values are recorded in dB which is a logarithmic scale, meaning that when there is a difference greater than a factor of 10, the difference in strength is by an order of magnitude.

To evaluate the variability in our recordings, we calculated the standard errors for the 40 kHz continuous tone tests displayed as error bars (Fig 6). Because analysis of the prey generated sounds does not target specific frequencies but needs to compare the signal response for a broad range of frequencies, we calculated a set of descriptive statistics for the average amplitude (dB) of each frequency for the five replicate recordings for all 10 models at each of the five distances. We then calculated descriptive statistics for the the standard deviation of those averages. The values for those calculations are available in the supplemental data (S2 Supporting Information).

A one-tailed student t test was used to demonstrate whether there were significant differences in signal strength (amplitude in dB) with the 40 kHz continuous tone recordings from the smooth model to the models with arrays (S1 Supporting information). We used a cut-off of 0.05 as the criterium for statistical significance.

## 3. Results

From the t-tests we found that 183 of 225 (91.1%) array signals were significantly different from the smooth model (40 kHz Supplementary Data) when comparing the 40 kHz continuous tone signal response of the nine array models to the smooth model for each of the five harmonics at the five different distances.

In relation to variation linked to the amplitudes produced from the recordings, there was consistently little variation in amplitude among the five replicate recordings for the 40 kHz continuous tone tests (S1 Supporting information) and the manufactured prey sound tests (S2 Supporting Information).

### 3.1 Hypothesis 1: The presence of a groove array will provide different acoustic information than that of a smooth surface

The signal responses from the grooved models show higher amplitudes at many of the harmonic frequencies compared to the smooth model as well as differences in response quality. Most of the harmonic frequencies of the 40 kHz continuous tone have greater amplitudes in the grooved pinna models than in the smooth model at all distances from 10 cm to 160 cm (Fig 6). For comparing the array models to that of the smooth model, we calculate differences of the dB amplitudes by subtracting the values from the array models from the smooth model and then graph the results (Fig 6). The values show that the arrays are predominantly more intense indicating which model better amplifies the signal. When combining all responses at all distances 175 of 225 values (77.8%) of the array amplitudes are stronger than the smooth model. The influence of the arrays appears to begin to wane at the 80 cm and 160 cm distance (Fig 6). If we look at the values leading up to 80 cm, we can see that 151 of 180 (~84%) of the array amplitudes are stronger than the smooth model (Fig 6, S1 Fig in S1 Supporting information 40 KHZ SUPPLEMANTARY DATA 40 KHZ SUPPLEMENTARY DATA).

We illustrate differences in response quality with the prey sounds test results. Spectrograms, power spectra, and frequency difference graphs show that the signal intensity is greater at higher frequencies (50–150 kHz) in the grooved pinna models than in the smooth model (Fig 7, S2 Supporting information). The 1mm5g model spectrogram was most like the smooth model spectrogram; the 3mm10g model spectrogram was most different from the smooth model spectrogram with the strongest differences occurring between the 3mm10G and the smooth models at 10 cm. In Fig 7C, the smooth and array model swap back and forth until about 60 kHz where the array differences begin to remain more intense in signal strength which can be seen up to 10 dB at 60 kHz and again at 85 kHz; minor differences (>2 dB) occur as high as 140 kHz (S1 Supporting information).

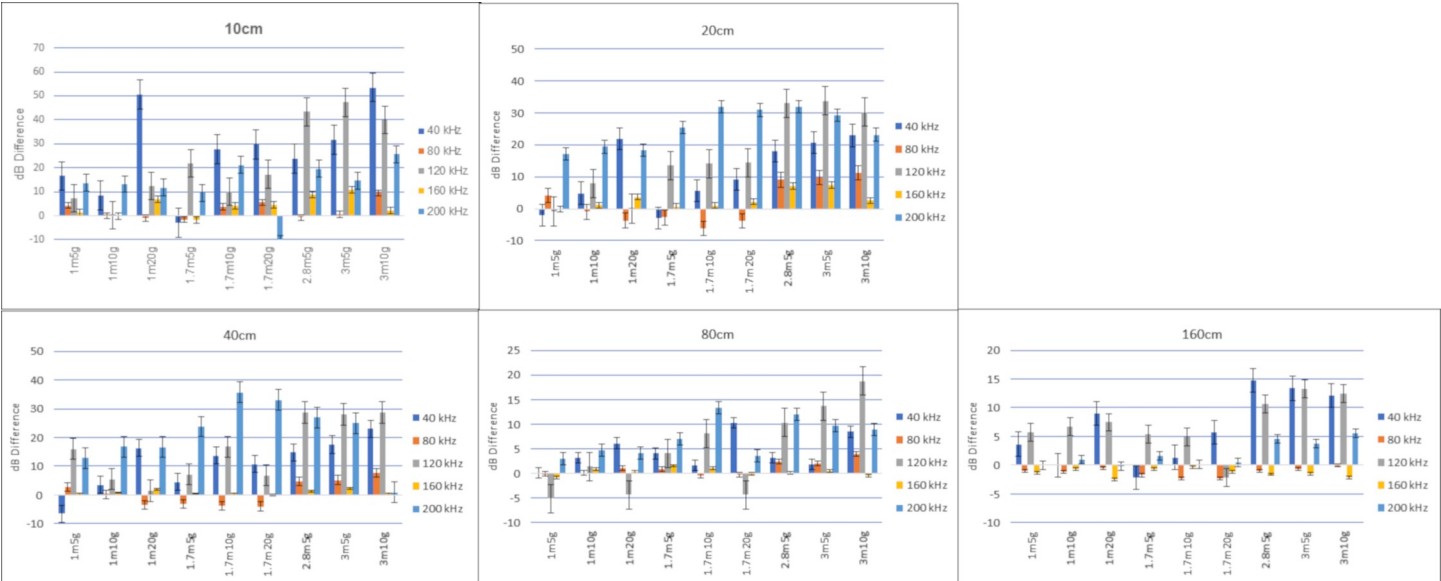

**Fig 6. Signal response difference graphs of the smooth model compared to the nine models with grooves.** Amplitude in decibels (dB) at 5 different frequencies comparing 9 different pinna models (see Table 1) to that of the smooth model. Each model was exposed to a 40 kHz continuous tone with five harmonics at five different distances. When smooth model amplitudes are more intense, they drop below zero.

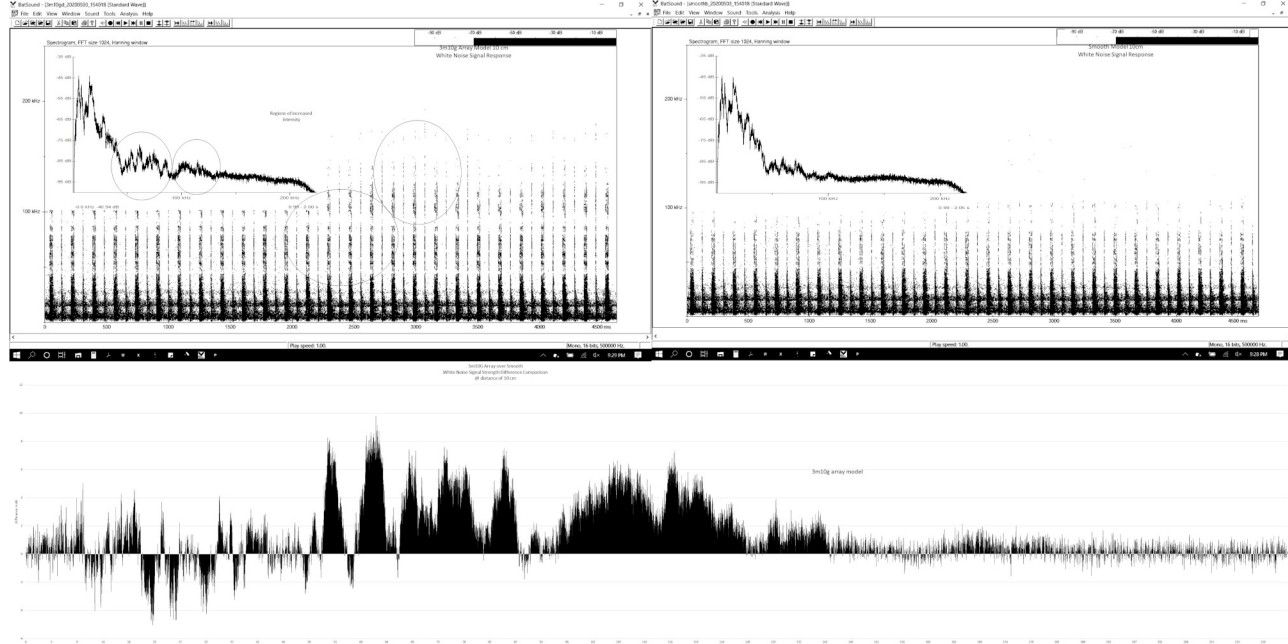

**Fig 7. Spectrograms, power spectra and a difference graph comparing the 3mm10G pinna model and the smooth pinna model when exposed to a sound that mimics prey rustling in leaves.** Spectrograms and power spectra of the 3mm10G pinna model (A) and the smooth pinna model (B) exposed to manufactured prey sounds at a distance of 20 cm. The circled areas indicate amplitude differences between the models. The frequency difference graph (C) plots the frequency (kHz) differences in amplitude (dB) between the grooved and the smooth model. Amplitudes above the 0 dB line are stronger in the 3mm10g model; amplitudes below the 0 DB line are stronger in the smooth model.

### 3.2 Hypothesis 2: Based on the principles of diffraction, the presence of a groove array will alter the acoustic signal based on the array's characteristics

We can see from Fig 6 that each model uniquely influences the signal. The same thing happens when comparing the signal response from each model when contacted by the prey generated sounds (S2 Supporting Information).

### 3.3 Hypothesis 2a: Amplitude will increase as the number of grooves increase

In the results of the 40 kHz continuous tone tests, we did not see a relationship between the numbers of grooves and the amplitude of the reflected sound waves (Fig 8, S1 Supporting information). We therefore cannot accept our hypothesis that amplitudes of soundwaves will be higher when reflected by pinna with greater number of grooves.

### 3.4 Hypothesis 2b: Wavelengths that correspond to specific groove separation widths will have a marked increase in amplitudes

The results of the 40 kHz continuous tone tests indicate that the amplitude of wavelengths that correspond to groove separation is increased by the grooved pinna models (Fig 9). From Table 1, a groove with a separation of 1mm corresponds to 343 kHz, a 1.715 mm groove to 200 kHz, a 2.858 mm groove to 120 kHz, and a 3 mm groove to 114.33 kHz.

The amplitude intensity is stronger for wavelengths matching 200 kHz reflected out of a groove array with 1.7 mm separations than when reflected by groove arrays with groove separations that do not match the wavelength (Fig 9). Similarly, the amplitude intensity is stronger for wavelengths that more closely match 120 kHz such as the models with 2.8 mm and 3 mm separations than the models with groove arrays that do not match the wavelength as closely. Pinna models with 1 mm groove widths do not show a consistent response to any frequency. Consequently, we can accept our hypothesis that wavelengths reflected by grooves with corresponding widths do increase in amplitude.

## 4. Discussion

Our tests showed that the presence of groove arrays increase the signal intensity compared to the pinna model without grooves. The 40 kHz continuous tone data shows that the majority (83.8% n = 180) of all harmonics generated by the arrays up to 80cm result in stronger signals than those of the smooth model (S1 Supporting information). But at 160 cm the influence drops to 53% (n = 45) of the signals from array models being stronger than the smooth model (S1 Supporting information).

Groove arrays influence the signal by increasing the amplitude of select frequencies within the spectral envelope which is dependent on the array configuration. The largest groove separations of 3 mm and 2.8 mm have the strongest influence, and the different array configurations alter the signals mostly as expected in relation to the principles of diffraction and wave behavior but differ from the predominant studies on diffraction gratings. The single strongest difference in amplitude occurred between the smooth model and the 1.715 mm 10 groove model at 40 cm resulting in a signal that was 32 dB more intense which was 34% stronger (1.715mm 10 groove = -57.7 dB; smooth = -89.8 dB). However, the strongest change in proportion occurred between the smooth and the 3 mm width 10 groove model at a distance of 10 cm resulting in a signal that was only 4 dB more intense but was 53% stronger (3mm10g = -3.5 dB; smooth = -7.5 dB).

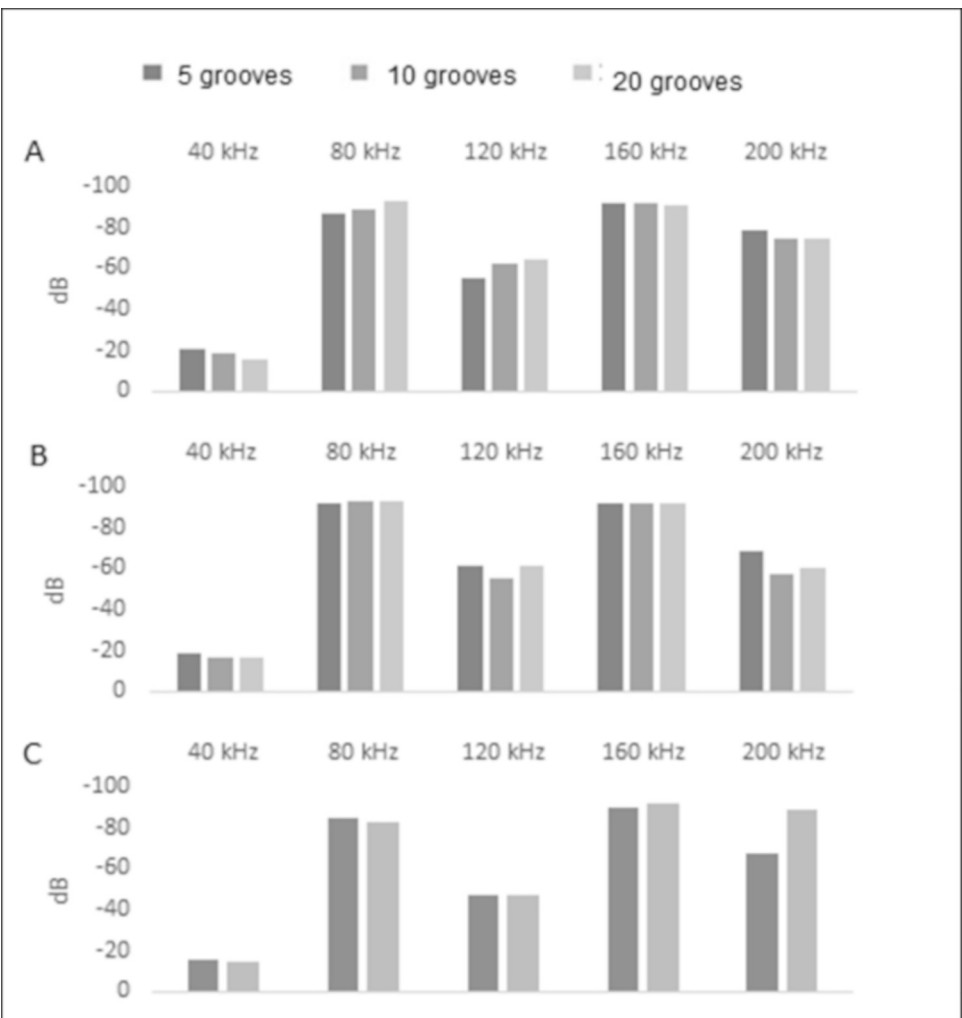

**Fig 8. Groove number amplitude comparisons from the 40 kHz continuous tests.** Amplitude in decibels (dB) of 40 kHz continuous tone sound waves with harmonics generated at 40 cm distance and reflected by pinna models with 5,10, and 20 grooves. Groove separation was 1 mm (A), 1.7 mm (B), and 3 mm (C). The strongest signal responses denoted in dB are closer to zero.

The influence of the arrays appears to diminish as the distance increases with the 40 kHz continuous sound source. We conclude that the acoustic effects caused by groove arrays does increase the signal strength and therefore would improve detectability and because the response by the acoustic signal varies according to the array characteristics, the array structures in the pinnae of animals are likely linked to their biological needs.

There was little evidence that the groove number affected the signal consistent with descriptions from traditional grating studies [8]. We are not sure why increasing the groove number does not follow what is typically found in other diffraction grating studies. Our study primarily focused on ascertaining if and not how the arrays change the signal. However, there are possible differences in our study design that could affect the performance when comparing to more traditional diffraction grating studies which are typically applied to optical wavelengths using fixed planar (flat) gratings with slit or groove widths that are much larger than the wavelengths being evaluated [8].

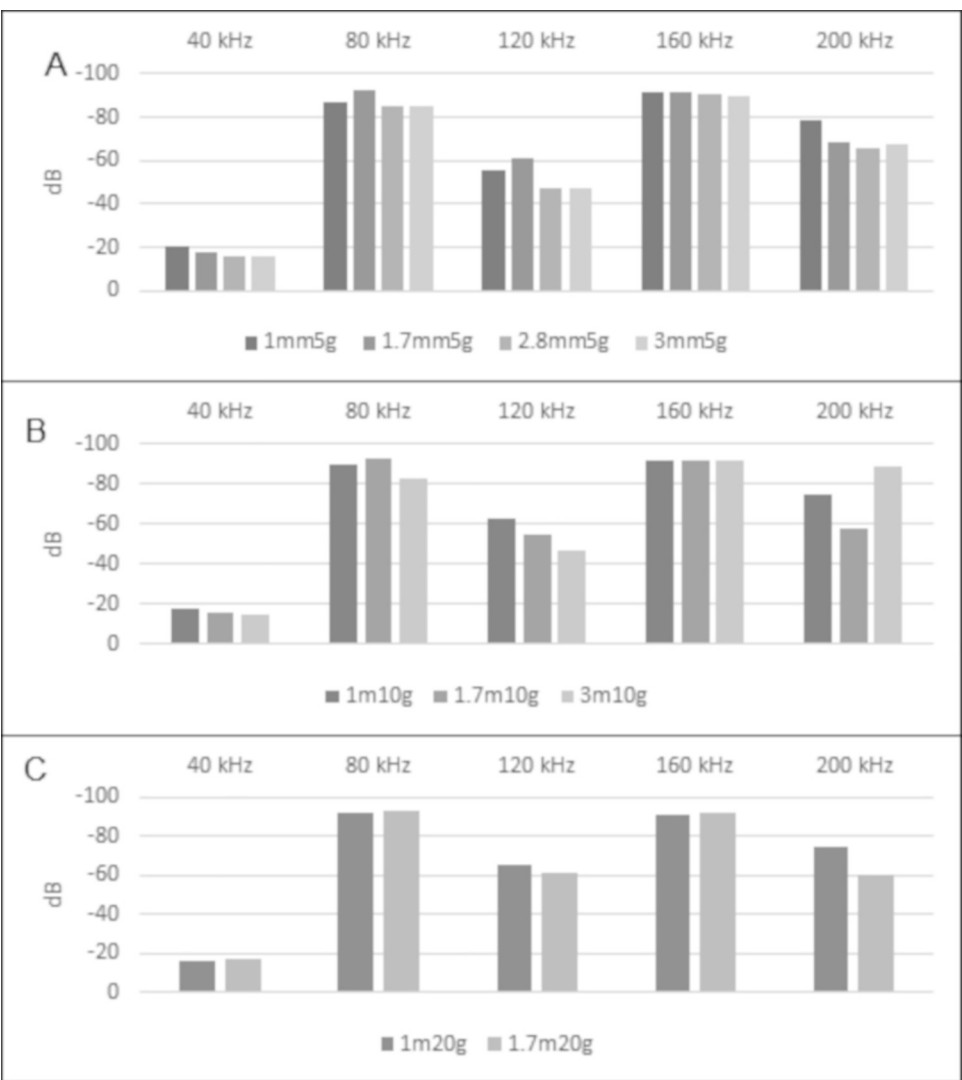

**Fig 9. Groove width amplitude comparisons from the 40 kHz continuous tests.** Amplitude in decibels (dB) of 40kHz continuous tone sound waves with harmonics generated at 40 cm distance and reflected by pinna models with groove separations of 1 mm, 1.7 mm, 2.8 mm, and 3 mm for pinna models with 5 grooves (A), 10 grooves (B), and 20 grooves (C). The strongest signal responses denoted in dB are closer to zero.

For example, gratings in our models and in the ears of animals differ from traditional optical grating studies because the grooves are not planar but are horizontally curved creating a concave reflective surface which would act to focus the signal (Fig 1). Optical studies typically use fixed sources and fixed gratings, whereas animals can move their heads and ears with great precision adjusting the array to fine tune the incoming signal, and many animals do have the ability to change the shape of the ears and arrays which likely affects the signal strength and content that reaches the tympanum. Other differences include that the wavelengths involved in this study sometimes straddle the array groove widths with the fundamental and some of the harmonics being smaller, larger or equal to the widths involved.

In addition, most optical diffraction studies use gratings with widths of 700–100 micrometers, which are much larger than the visible light wavelengths they diffract (400–700 nanometers) [8]. In our models, the strongest responses come from arrays with groove widths of 2.8

and 3 mm which are being contacted by the 40 kHz fundamental (λ = 8.5 mm), the second harmonic of 80 kHz (λ = 4.3 mm) and the third harmonic of 120 kHz (λ = 2.8 mm) whose wavelengths all surpass the 3 mm and matches the 2.8mm groove widths. The 4th and 5th harmonics (160 kHz = λ 2.1mm, 200 kHz = λ 1.7 mm respectively) are much smaller than both of the 2.8 mm and 3 mm groove widths. In other words, the results of studies done on optical gratings is not directly informative to the results of this study. Most acoustically reliant animals can be observed continually adjusting their head and/or the pinnae while listening for incoming sounds. Since our models were conducted using just one array (not paired as in most animals) and held static during recording, we likely could not capture the full extent of the information produced by the arrays like those operated by a live animal. It is also worth noting that most echolocating bats do have a tragus and that there is likely a correlation to array variability and the positioning, shape and structure of the tragus. In owls, tests will need to determine if the operculum serves a similar function as mobile pinnae and/or the tragus in mammals.

Many animals that passively listen for sounds often have specialized sound gathering structures in the form of enlarged ears [2,3], or in the case of some owls, a facial disc or ruff [19,20]. In mammals and owls, the presence of arrays in the ears appears to be linked to nocturnal behavior where low light conditions tend to increase the value of acoustic information. Our tests using the prey generated sounds show that in comparison to the model with the smooth surface, the array configuration predominantly affects frequencies ranging from 3 to 120 kHz extending to a lesser degree up to 143 kHz (Fig 7).

Although the hearing ranges of the mammal species with arrays at the ear openings [4] are largely unknown, the hearing of the few bats that have been studied are quite variable, extending from as low as 2 kHz to just above 100 kHz [23,24]. But probably more important would be to learn which frequencies fall into sensitivity peaks or the "best" hearing range for a particular species. The peak hearing ranges for the bats that have been studied appear to be clustered from 15–80 kHz depending on the species [23,24]. When we look at the broadband signal response from the prey generated sounds, we can see that there is a consistent peak in signal strength favoring the arrays over the smooth model within that 15–80 kHz best hearing range (Fig 7, S2 Supporting Information).

A select group of owls with facial discs do have groove arrays associated with their auricular openings. The hearing range for most owls is undetermined. However, the barn owl (*Tyto alba*) has been extensively studied and reportedly has a hearing range from 200 Hz to 10 kHz with the optimal hearing range between 3–9 kHz [19,21,22]. We did not record precise measurements of the groove arrays in the owls, but it appears that the groove width in those we examined were approximately 1 mm and had more than 10 grooves (Fig 4). The 3–9 kHz range signal responses of the 1 mm 10 and 20 groove arrays are generally stronger when compared to that of the smooth model using the prey generated sounds (S2 Supporting Information). Although inconsistent, it does appear that the arrays in our models are able to increase the signal strength for wavelengths within the optimal hearing range of the barn owl. Knowing what we now know about how arrays affect acoustic signals and that arrays are present at the auricular openings of owls; further studies are warranted. It could be worthwhile to know specifically which owl species have arrays and how the configuration of those arrays may be affecting the sounds owls hear.

Physical principles tell us, and direct observations show us that structural features affect wave behavior. Our study suggests that the strength of a reflected acoustic signal that contacts an array will be consistently stronger over those that contact a smooth surface, and that the signal response does depend on the frequencies in relation to the array configuration. Although we cannot yet predict exactly how the array configuration will alter the signal, it should be

something that can be calculated. To really know if an array has value to a given species, the acoustic signal response from the array configuration in the ear of that animal should be compared to that species best hearing range and the acoustic frequencies they encounter. If the array is in fact increasing the signal gain for those best hearing frequencies, then it can be assumed the array is beneficial to that animal.

## 5. Conclusion

This study provides compelling evidence that the primary function of groove arrays associated with auricular openings is to strengthen the acoustic signal and that the array characteristics will uniquely influence the signal. We noted differences >32 dB stronger in amplitude from the arrays over that of the smooth model and many examples where the amplitude was greater than 10 dB. When the presence of an array changes the amplitude of an incoming sound by orders of magnitude it is likely to be an evolutionary advantage, especially if it can increase listening distances, enhance the ability to pinpoint source location or provide important details. However, a substantial list of unanswered questions has emerged on how the array characteristics uniquely affect the signal and how the variations may contribute to the information needed to interpret incoming signals. Among the animals with groove arrays we reviewed, why is there such a wide variety of groove widths if the largest widths provide the strongest effects? For mammals where the grooves are formed from innervated muscles, if the animal can modify the signal content by physically altering the array characteristics (i.e. groove width) it seems highly advantageous. How does the tragus on a bat or the operculum on an owl interact with the array characteristics to influence the signal? How do the array dimensions and position within the reflective surface affect the signal? Is it possible to determine if the arrays can affect wavelengths that straddle the conventional array widths? How sensitive are the arrays to adjustments to the angle of incidence? How would an animal like a bat with large ears control an arrays angle of incidence during flight to account for aerodynamics? This is a short list of questions, but as we begin to understand the value of the arrays, answers to some of these questions should help further resolve what kind of information is being generated and how the animals might be applying the many unique array variations to their specific biological needs.

## Supporting information

**S1 Supporting information. 40kHz continuous tone averages and difference graphs.**
(XLSX)

**S2 Supporting information. Prey generated sounds difference graphs.**
(XLSX)

**S3 Supporting information. 40 kHz continuous tone tests raw data.**
(XLSX)

**S4 Supporting information. Prey generated sounds raw data.**
(XLSX)

## Acknowledgments

This study was made possible through continuous consultation with Lars Pettersson of Pettersson Elektroniks, early communication from Roman Kuc and Ed West provided consultation on content and research design. We thank Corky Quirk of NorCal Bats for allowing observations of captive live bats, the Museum of UC Davis, California for access to review bat and owl

specimens, Steven Lucero for his patience and advice with model designs, Sherri and Brock Fenton for access to bat photographs and to the Witmer Lab at Ohio University in collaboration with the University of Texas at Austin. for access to the barn owl figure.

## Author Contributions

**Conceptualization:** Brian W. Keeley, Annika T. H. Keeley.

**Data curation:** Brian W. Keeley.

**Formal analysis:** Brian W. Keeley.

**Investigation:** Brian W. Keeley.

**Methodology:** Brian W. Keeley, Annika T. H. Keeley.

**Project administration:** Brian W. Keeley.

**Resources:** Annika T. H. Keeley.

**Supervision:** Brian W. Keeley.

**Validation:** Brian W. Keeley.

**Visualization:** Brian W. Keeley, Annika T. H. Keeley.

**Writing – original draft:** Brian W. Keeley, Annika T. H. Keeley.

**Writing – review & editing:** Brian W. Keeley, Annika T. H. Keeley.

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
