## [Decision Letter · Decision Letter 0]

23 Jul 2021

PONE-D-21-14020

Groove arrays in the ears of mammals and owls enhance acoustic information.

PLOS ONE

Dear Dr. Keeley,

Thank you for submitting your manuscript to PLOS ONE. After careful consideration, we feel that it has merit but does not fully meet PLOS ONE’s publication criteria as it currently stands. Therefore, we invite you to submit a revised version of the manuscript that addresses the points raised during the review process.

In particular, it is important you address the issue of multiple repetitions of the measurements. This would be needed to ensure the results are statistically sound. Further discussing the relevance of your measurements to real signals perceived by bats and owls would also be useful, given the title of your paper.

We look forward to receiving your revised manuscript.

Kind regards,

Vivek Nityananda

Academic Editor

PLOS ONE

Reviewers' comments:

Reviewer's Responses to Questions

**Comments to the Author**

1. Is the manuscript technically sound, and do the data support the conclusions?

Reviewer #1: Partly

Reviewer #2: Yes

Reviewer #3: Partly

2. Has the statistical analysis been performed appropriately and rigorously? 

Reviewer #1: No

Reviewer #2: Yes

Reviewer #3: N/A

3. Have the authors made all data underlying the findings in their manuscript fully available?

Reviewer #1: Yes

Reviewer #2: Yes

Reviewer #3: No

4. Is the manuscript presented in an intelligible fashion and written in standard English?

Reviewer #1: Yes

Reviewer #2: Yes

Reviewer #3: Yes

5. Review Comments to the Author

Reviewer #1: This is an interesting experimental investigation into the role of grooves patterns that can be found in bat ears. However, if I understand the description of the methods and the results correctly, the comparisons between grooved and smoothed shapes were done using only single acoustic measurement for each shape. In practice, acoustic measurements and spectrum estimates based on them are subject to a very high level of variability. It is hence absolutely essential that the authors add multiple repetitions to their measurements and investigate which of the observed differences are statistically significant. Without such an analysis, it is impossible to draw any conclusions from the obtained results.

In addition, I believe that it is essential for the authors to remove all mentions of "acoustic information" from their hypotheses since there is nothing in their work/analysis that could be seen as measuring information.

Finally, the authors should comment on how the studied frequency range compares to the frequency bands known to be utilized by bats and owls.

Here some minor issues I have come across:

l. 47: There seems to be no benefit to using the term "polyphonic"

here, which afaik means independent melodies, not just multiple

frequencies in music .

l. 82: Would say "phase relationships" to avoid confusion with medical

use of phasic (i.e., short-term).

l. 83: Use "broadband" instead of "polyphonic".

l. 123: It is not informative to say that the pinna models were based

on "discussions with acoustic specialists", instead the specialists

should be thanked in the acknowledgments.

l. 134: The "z" in "kHz" is not capitalized.

l. 150: More information about the prey noise should be given, e.g.,

bandwidth, spectral shape (flat, lowpass, bandpass characteristics).

l. 196 - 198: It appears that no repetitions of the expriments

l. 340: Capitalize "B" in "dB".

l. 359: Remove comma after "Kuc".

Reviewer #2: This paper presents that the groove arrays on the ear model increase the signal intensity compared to the smooth model. It provides fundamental study for many further open questions. There are a few concerns that could be addressed to improve the quality of the paper.

1. This paper mainly investigates the effect of groove array on the general ear model instead of the animal’s ear. I’d recommend that the Short Title is more appropriate than the Full Title.

2. A setup figure (or a sketch of the emitter, ball, sandpaper discs, etc) shows how those three sound sources are generated will help the reader to understand.

3. Including example spectrograms of array model and smooth model in Fig 6 could help the reader to understand how you get the dB difference plots.

4. Please make sure all figures have x-axis and y-axis labels/units and suitable font sizes. In fig 8, the time duration (1000 ms) in the x-axis should be consistent for all subplots.

5. In line 50 “Huygens-Fresnell principle” should be “Huygens-Fresnel principle”.

In line 250 “...intense. in signal strength...”.

In line 338 “db” should be “dB”.

6. In line 131, is there any specific reason to choose the ear model dimensions to be 28 mm X 44 mm?

7. Is the statement in lines 308-309 based on the results from fig 6? If so, it is different from the statement of lines

236-237.

8. The statement in line 314: “The influence of the arrays appears to diminish as the distance increases with one sound

source but not another.” Replace the “one sound source, another” with the specific name of the sound source to

avoid ambiguity.

9. In the “Discussion” section, the author mentioned “traditional diffraction grating studies” many times and compared

their results with it. References are needed to the “traditional diffraction grating studies”.

10. Did the prey sound test involve 10 models and 5 distances as well? If so, any further statistical analysis could be

put into the supplementary material. Currently, only one fig (fig 7) shows one condition.

11. In lines 337-338: “This study demonstrates that the presence of arrays appears to be able to enhance the signal

strength across the spectral envelope, but the effective distance may vary according to the sound source.”

The “sound source” here indicates 3 different types of sound sources that are continually mentioned in the paper or

different frequency sounds from the "40kHz sound source"? It should be stated more clearly.

Reviewer #3: There are two major issues with this paper.

1. The ridges do much better than the pinna without ridges, particularly the increase in gain. Could there be a problem with that pinna that evolution has solved using thin flexible skin?

2. More important than 1: The paper gives quantitative and qualitative results that indicate the effect of the ridges, but little analysis or explanation on how they would benefit a bat. The experiments and ridge designs were too exhaustive in scope. This reviewer would have liked to see one or two signals with the best ridge design, in the author's opinion, to demonstrate the benefit of the ridges, beyond that the ridges modify the detected signals.

6. PLOS authors have the option to publish the peer review history of their article (what does this mean?). If published, this will include your full peer review and any attached files.

Reviewer #1: No

Reviewer #2: **Yes: **Xiaoyan Yin

Reviewer #3: No

---

## [Author Response · Author response to Decision Letter 0]

7 Sep 2021

We have uploaded a "Response to Reviewers" letter as requested that addresses each comment.

---

## [Decision Letter · Decision Letter 1]

28 Sep 2021

PONE-D-21-14020R1Acoustic wave response to groove arrays in model ears.PLOS ONE

Dear Dr. Keeley,

Thank you for submitting your manuscript to PLOS ONE. After careful consideration, we feel that it has merit but does not fully meet PLOS ONE’s publication criteria as it currently stands. Therefore, we invite you to submit a revised version of the manuscript that addresses the points raised during the review process.

In particular, it would be important to clarify the point raised by the reviewer about what exactly the variability in the measurements entails.==============================

We look forward to receiving your revised manuscript.

Kind regards,

Vivek Nityananda

Academic Editor

PLOS ONE

Journal Requirements:

Additional Editor Comments (if provided):

Reviewers' comments:

Reviewer's Responses to Questions

**Comments to the Author**

1. If the authors have adequately addressed your comments raised in a previous round of review and you feel that this manuscript is now acceptable for publication, you may indicate that here to bypass the “Comments to the Author” section, enter your conflict of interest statement in the “Confidential to Editor” section, and submit your "Accept" recommendation.

Reviewer #1: (No Response)

Reviewer #3: All comments have been addressed

2. Is the manuscript technically sound, and do the data support the conclusions?

Reviewer #1: Partly

Reviewer #3: Yes

3. Has the statistical analysis been performed appropriately and rigorously? 

Reviewer #1: I Don't Know

Reviewer #3: N/A

4. Have the authors made all data underlying the findings in their manuscript fully available?

Reviewer #1: Yes

Reviewer #3: Yes

5. Is the manuscript presented in an intelligible fashion and written in standard English?

Reviewer #1: No

Reviewer #3: Yes

6. Review Comments to the Author

Reviewer #1: The revision address all my concerns with the exception of the variability in the measurements. The manuscript should clarify whether 5 measurements were made for each of the 5 different distances, or just one measurement was made per distance. In the former case, error bars (e.g., max and min differences found) should be included in Fig. 6 to show the variability of the measurements. If only 1 measurement per distance has been made been made, additional measurements should be done to assess the variability in these experimental results.

Reviewer #3: (No Response)

7. PLOS authors have the option to publish the peer review history of their article (what does this mean?). If published, this will include your full peer review and any attached files.

Reviewer #1: No

Reviewer #3: No

---

## [Author Response · Author response to Decision Letter 1]

21 Oct 2021

We have addressed the suggestions you provided. Thank you for investing your knowledge in improving the study results and thank you for your persistence.

---

## [Editor Report · Decision Letter 2]

2 Nov 2021

Acoustic wave response to groove arrays in model ears.

PONE-D-21-14020R2

Dear Dr. Keeley,

We’re pleased to inform you that your manuscript has been judged scientifically suitable for publication and will be formally accepted for publication once it meets all outstanding technical requirements.

Kind regards,

Vivek Nityananda

Academic Editor

PLOS ONE
---

## [Editor Report · Acceptance letter]

5 Nov 2021

PONE-D-21-14020R2 

Acoustic wave response to groove arrays in model ears. 

Dear Dr. Keeley:

I'm pleased to inform you that your manuscript has been deemed suitable for publication in PLOS ONE. Congratulations! Your manuscript is now with our production department. 

Kind regards, 

on behalf of

Dr. Vivek Nityananda 

Academic Editor

PLOS ONE